# Depth-Specific Soil Electrical Conductivity and NDVI Elucidate Salinity Effects on Crop Development in Reclaimed Marsh Soils

José Luis Gómez Flores [1], Mario Ramos Rodríguez [1], Alfonso González Jiménez [1], Mohammad Farzamian [2], Juan Francisco Herencia Galán [3], Benito Salvatierra Bellido [4], Pedro Cermeño Sacristan [3] and Karl Vanderlinden [1,*]

[1] IFAPA Centro Alameda del Obispo, 14004 Córdoba, Spain; josel.gomez@juntadeandalucia.es (J.L.G.F.); mario.ramos.r@juntadeandalucia.es (M.R.R.); trabajoaj303@gmail.com (A.G.J.)

[2] Instituto Nacional de Investigação Agrária e Veterinária, 2780-157 Oeiras, Portugal; mohammadfarzamian@gmail.com

[3] IFAPA Centro Las Torres, 41200 Alcalá del Río, Spain; juanf.herencia@juntadeandalucia.es (J.F.H.G.); pedro.cermeno@juntadeandalucia.es (P.C.S.)

[4] FAPA Centro Rancho de la Merced, 11431 Jeréz de la Frontera, Spain; benito.salvatierra@juntadeandalucia.es

[*] Correspondence: karl.vanderlinden@juntadeandalucia.es

**Abstract:** Agricultural management decision-making in salinization-prone environments requires efficient soil salinity monitoring methods. This is the case in the B-XII irrigation district in SW Spain, a heavy clay reclaimed marsh area where a shallow saline water table and intensively irrigated agriculture create a fragile balance between salt accumulation and leaching in the root zone, which might be disrupted by the introduction of new crops and increasing climate variability. We evaluated the potential of electromagnetic induction (EMI) tomography for field-scale soil salinity assessment in this hyper-conductive environment, using EMI and limited analytical soil data measured in 2017 and 2020 under a processing tomato–cotton–sugar beet crop rotation. Salinity effects on crop development were assessed by comparing Sentinel 2 NDVI imagery with inverted depth-specific electrical conductivity (EC). Average apparent electrical conductivity (ECa) for the 1-m depth signal was 20% smaller in 2020 than in 2017, although the spatial ECa pattern was similar for both years. Inverted depth-specific EC showed a strong correlation (R ≈ 0.90) with saturated paste extract EC (ECe), [Na$^+$] and sodium absorption ratio (SAR), resulting in linear calibration equations with R$^2$ ≈ 0.8 for both years and leave-one-out cross validation Nash–Sutcliffe Efficiency Coefficient, ranging from 0.57 to 0.74. Overall, the chemical parameter estimation improved with depth and soil wetness (2017), yielding 0.83 < R < 0.98 at 0.9 m. The observed spatial EC distributions showed a steadily increasing inverse correlation with NDVI during the growing season, particularly for processing tomato and cotton, reaching R values of −0.71 and −0.85, respectively. These results confirm the potential of EMI tomography for mapping and monitoring soil salinity in the B-XII irrigation district, while it allows, in combination with NDVI imagery, a detailed spatial assessment of soil salinity impacts on crop development throughout the growing season. Contrary to the popular belief among farmers in the area, and despite non-saline topsoil conditions, spatial EC and subsoil salinity patterns were found to affect crop development negatively in the studied field.

**Keywords:** apparent electric conductivity; B-XII irrigation district; NDVI; soil salinity; reclaimed marsh soils

## 1. Introduction

Soil salinization represents a latent threat to soil quality and agricultural sustainability in the regions where proper agricultural management in response to specific environmental conditions has enabled the development of a fragile balance between salt buildup and removal rates in the soil profile. Potentially changing climate conditions, short-term fluctuations in irrigation water availability and quality, and the saline water table depth, or

changes in the soil and water management might disrupt this equilibrium with negative consequences for crop production and soil functionality [1,2].

To keep track of the soil salinity status in such environments, a growing demand for efficient field monitoring methods exists. Yet, conventional soil monitoring entails periodical soil sampling and laboratory analysis [3], which are time-consuming, labor-intensive and expensive. In addition, the measurements cannot be repeated at the same locations since the soil sampling procedure is destructive. Alternatively, networks of permanently installed electromagnetic sensors can be used to measure the soil moisture, temperature and bulk electrical conductivity at fixed depths [4]. Although such an approach yields quasi-continuous measurements in time, it provides only limited spatial information at the locations where the sensors are installed.

Detailed spatial soil information can be obtained through electromagnetic induction (EMI) sensing, which has become one of the most popular methods for characterizing the spatial variability of soils and their properties and states at the field scale [5,6], since it is fast and easy to deploy in the field, mainly due to its non-contact and non-invasive nature and its large measurement support volumes ($\sim m^3$). This technique allows the simultaneous measurement of the integrated apparent electrical conductivity (ECa), measured across different soil depths. Under non-saline soil conditions, ECa is usually related to clay and soil water content, among other soil properties, while under saline conditions it is the contribution of the solute concentration of the soil water that dominates the ECa signal. When integrated in a mobile measurement platform, these instruments can scan large areas and take thousands of measurements within a couple of hours. The georeferenced ECa data are then mapped and related to independent measurements of the relevant soil properties for calibration [7–9].

Yet, this approach does not provide information on the vertical distribution of the "true" soil conductivity (EC) and the related soil properties. The recent methodological advances in hydrogeophysics [10] are unlocking the full potential of EMI through the joint inversion of multi-receiver data [11,12], by estimating the vertical distribution of EC across the soil profile, from which the soil salinity profiles can be estimated in 2D and quasi-3D [13–17].

Once the spatial distribution of the soil salinity is known, its effect on crop development can be assessed, using remote sensing imagery. Since ECa is generally an indicator of soil fertility [5] most of the studies show positive relationships between NDVI [18] and ECa [19]. However, if ECa is correlated with soil conditions that constrain or affect the crop development negatively, then inverse relationships can be expected [20].

In this work we evaluate the potential of EMI tomography for mapping and monitoring soil salinity between 2017 and 2020 in a commercial field in the B-XII irrigation district in SW Spain. In addition, the use of Sentinel 2 NDVI imagery for assessing the impact of soil salinity on crop development is evaluated. The specific objectives are: (1) to assess the spatial variability of ECa and inverted EC in 2017 and 2020; (2) to calibrate the estimated EC images using minimal soil analysis data and estimate the saturated paste extract conductivity (ECe) and sodium absorption ratio (SAR); and (3) to evaluate the potential of the combined use of ECa and NDVI imagery for assessing the impact of soil salinity on crop development in the B-XII irrigation district.

## 2. Materials and Methods

### 2.1. Field Description

This study was conducted in a 4-ha commercial field in the reclaimed saline marsh soils of the B-XII irrigation district (Lebrija, Spain; Figure 1). The rather homogeneous heavy clay soil, classified as Gleyic Fluvisol [21], is underlaid by an artificially drained shallow saline water table [22,23]. The drainage system in the field consisted of 0.3-m long ceramic sections that form parallel 250-m long pipes, buried at a depth of 1 m and separated by 10 m. The pipes discharged into a drainage ditch along the northern limit of the field.

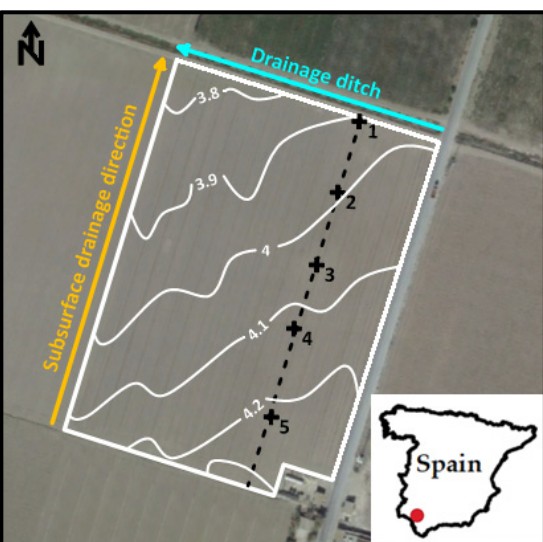

**Figure 1.** Image of the study field with topography and location of the five sampling points along the transect (Google Earth, 2022) [24].

The climate is typically Mediterranean, with moderate humid winters and hot dry summers. The annual rainfall ranges from 350 to 1100 mm, with an average of 550 mm (2000–2020). Most of the precipitation falls from November to March. There is practically no rainfall during July and August. The average annual reference evapotranspiration is about 1000 mm (years 2000–2020) [25]. The study period comprised a complete summer crop rotation, from 2017 to 2020, with processing tomato (*Solanum lycopersicum*), cotton (*Gossypium hirsutum* L.) and sugar beet (*Beta vulgaris*), respectively.

### 2.2. Electromagnetic Induction Sensing of ECa and Inversion

The ECa was measured in November 2017 and February 2020, using a DualEM-21S (Dualem Inc., Milton, ON, Canada). This instrument contains dual-geometry receivers (horizontal, HCP/ perpendicular, PRP) at 1 m (HCP1 and PRP1 signals) and 2 m (HCP2 and PRP2 signals) from the transmitter and allows for simultaneous conductivity sounding down to theoretical depths of exploration near 1.5, 0.5, 3 and 1 m (ECa-1.5, ECa-0.5, ECa-3, ECa-1), respectively. The sensor was operated at a height of 0.105 m above the soil surface in a customized polyvinyl chloride (PVC) sled towed by an all-terrain vehicle (ATV). A real-time kinematic differential global positioning system (Trimble, Sunnyvale, CA, USA) was used for georeferencing the ECa measurements. The ECa data and coordinates were logged on a mesa3 field computer (Juniper Systems, Logan, UT, USA). In 2017, the measurements were performed in the direction of the drainage pipes and in the perpendicular direction. In 2020, the measurements were only performed in the former direction, since the field was ridged and not transitable in the perpendicular direction. Due to hardware configuration problems, only the PRP2 (1 m) and HCP2 (3 m) signals were fully logged during the survey in 2017. The measurements were made at an average speed of 7 km h$^{-1}$ with a separation distance of 4–5 m between the adjacent passes, which provides an approximate spatial data density of 0.25 points m$^{-2}$.

Figure 2 shows the antecedent meteorological conditions for each ECa measurement date. During the 30 days preceding the measurement in 2017, the total rainfall was 85.4 mm, distributed over days 18 and 19. In 2020, the total rainfall was 72.4 mm, distributed between days 21 and 28, without exceeding 20 mm d$^{-1}$. Moreover, a decreasing trend in daily ETo was observed before the measurement in 2017, in contrast to the increasing ETo trend observed before the ECa measurement in 2020. As a result, wetter soil conditions are expected in 2017 as compared to 2020.

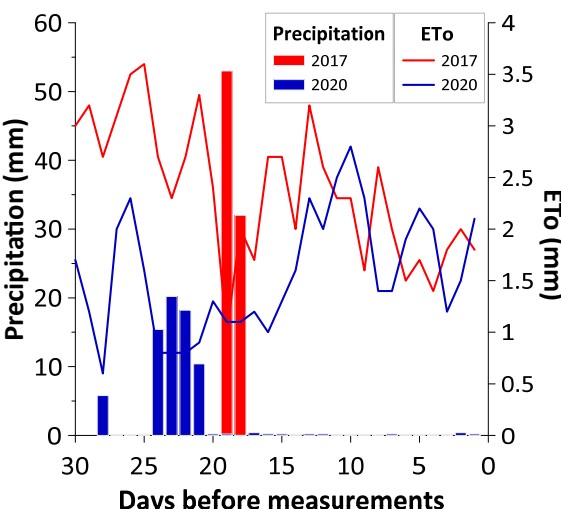

**Figure 2.** Daily rainfall and reference crop evapotranspiration (ETo) during the 30 days preceding the ECa measurements in 2017 and 2020.

The raw georeferenced ECa measurements were corrected for positional error caused by delays in the data transmission, reception or logging during the field measurement, according to the method proposed by [26]. The ECa data were interpolated on a 0.5 × 0.5 m grid using the inverse distance method, as implemented in Surfer (Golden Software, LLC Boulder, CO, USA), and the descriptive statistics were calculated from the interpolated data and compared for both of the surveys.

The interpolated values of the four EMI signals were inverted, using the EM4Soil software (EMTOMO, Lisbon, Portugal), to estimate the vertical distribution of the electrical conductivity (EC). The EM4Soil software estimates the bidimensional images of the EC distribution across the soil profile, conditioned on the neighboring ECa values. Given the high soil ECa in this area, the full solution was used. According to the imposed smoothness condition for the estimation of the EC profile, two inversion algorithms were considered (S1 and S2). Both are variations of the Occam regularization method [11]; the inversion algorithm, S2, was the one used in this study, since it produced smoother results than the S1. The 2D inversions were performed along a transect where the soil samples were taken (Figure 1; see also Section 2.3), while the 3D inversions were performed using all of the ECa data, from which field-wide depth-specific EC maps were obtained for 0.1, 0.3, 0.5, 0.7 and 0.9 m depths.

*2.3. Soil Sampling and Laboratory Analysis*

On both of the survey dates, soil samples were taken at five locations (P1, P2, ... P5) along a transect parallel to the drainage pipes (Figure 1), with 0.2-m depth increments down to 1 m, using a 0.05-m diameter Edelman soil auger. The sample depths are denoted as 0.1, 0.3, 0.5, 0.7 and 0.9 m and correspond to the depths at which the EC was estimated (see Section 2.1). The 25 soil samples obtained on each date were air dried, ground and passed through a 2 mm sieve. The saturated soil pastes were prepared by adding deionized water to 200 g of air-dried soil and allowing the sample to reach equilibrium during 24 h [27]. Subsequently, the extracts were collected and the ECe and pH were measured, using a conductivity/pH meter (Hanna Instruments, HI5521). The extracts were analyzed for $Na^+$, $K^+$, $Mg^{2+}$ and $Ca^{2+}$ concentrations, using an inductively coupled Plasma Optical Emission Spectrometry (ICP-OES) (Perkin Elmer Avio 200).

The sodium adsorption ratio (SAR) was computed as:

$$SAR = [Na^+]/[([Ca^{2+}] + [Mg^{2+}])/2]^{0.5} \ (mmol_c^{1/2} \ L^{-3/2}),$$ (1)

with cation concentrations in $mmol_c \ L^{-1}$ [28].

The chemical soil parameter (pHe, ECe, SAR, [Na$^+$], [Ca$^{2+}$], [Mg$^{2+}$] and [K$^+$]) data sets for 2017 and 2020 were subjected to a principal component analysis (PCAs) to infer the common patterns of variation [29]. In addition, the profile-averages and depth-specific spatial averages were compared for each year, using ANOVA and between years using Student *t*-tests. Finally, the depth-specific increments of soil properties between 2017 to 2020 ($\Delta$) were compared for the different sampling points

### 2.4. Estimation of Soil Salinity Status

The 2D inversion of the 2017 and 2020 ECa data was performed along the transect (Figure 1). The estimated EC values were then calibrated against the ECe and SAR, using linear regression models. The validation of the calibration models was performed using leave-one-out cross-validation (LOOCV), as implemented in the R "caret" package [30], using the Root Mean Squared Error (RMSE) as statistical evaluation parameters:

$$\text{RMSE} = \sqrt{\sum_{i=1}^{N} (X_{e,i} - X_{m,i})^2 / N};$$ (2)

the Mean Absolute Error (MAE):

$$\text{MAE} = \frac{1}{N} \sum_{i=1}^{N} \left| X_{e,i} - X_{m,i} \right|;$$ (3)

and the Nash–Sutcliffe Efficiency Coefficient (NSE),

$$\text{NSE} = 1 - \frac{\sum_{i=1}^{N} (X_{m,i} - X_{e,i})^2}{\sum_{i=1}^{N} (X_{m,i} - \overline{X}_{m,i})^2},$$ (4)

where $X_{e,i}$ and $X_{m,i}$ are the estimated and measured values, respectively, and $N$ is the number of measured and estimated data pairs. The NSE ranges from $-\infty$ to 1 and NSE > 0 indicates that the model is a better estimator than the measured mean [31]. For the interpretation of the NSE we used the criteria proposed by [32]. To produce the depth-specific ECe and SAR maps for 2017 and 2020, the field-wide depth-specific EC estimates obtained from the 3D inversion of the ECa data were used with the linear calibration equations obtained for ECe and SAR from the 2D inversions.

### 2.5. NDVI Imagery Processing and Analysis

The available Sentinel 2 NDVI imagery for the cropping seasons from 2017 to 2019 was analyzed using Google Earth Engine [33]. Thirty-six images, corresponding to the spring and summer periods when the field was cultivated and with less than 1% cloud cover, were retained for the analysis. Since the western part of the field remained fallow in 2017, an area of 1.7 ha (171 pixels) in the eastern part of the field was retained for the NDVI analysis of each cropping season. Before calculating the Pearson correlation coefficient, ECa and depth-specific EC were first interpolated on a grid with the same spatial resolution as the NDVI images (10 × 10 m). The spatiotemporal evolution of the NDVI of each crop along the study transect (Figure 2) was analyzed and the Pearson correlation coefficient with the depth-specific estimated ECe was calculated for each available NDVI image.

## 3. Results and Discussion

### 3.1. Soil Properties in 2017 and 2020

Overall, ECe, [Na$^+$] and SAR were significantly higher in the subsoil (0.5-0.9 m) than in the topsoil (0.1 and 0.3 m) for both of the survey years (Table 1), being 0.7 m the transition depth in 2017 for ECe and [Na$^+$] and 0.5 and 0.7 m the transition depth for SAR. In 2020, the transitions between the top- and subsoil properties were spread over a wider depth interval, ranging from 0.3 to 0.7 m for ECe and SAR, while the transition depth for [Na$^+$]

was again 0.7 m. The pHe, $[K^+]$, $[Mg^{2+}]$ nor $[Ca^{2+}]$ showed significant differences in depth, except for $[Mg^{2+}]$ which showed in 2020 a transition at 0.5 and 0.7 m between significantly lower values in the topsoil as compared to the 0.9 m layer. This vertical distribution pattern of the soil properties is in accordance with the presence of a shallow saline water table (EC~ 50 dS m$^{-1}$), which is generally kept below 1 m by the tile-drain system. However, salinization of the upper soil layers can occur during the summer growing season when the water table rises, because of intense irrigation to meet the maximum crop water demands and capillarity promotes the upward movement of the saline water towards the topsoil, where it evaporates or transpires under conditions of high atmospheric water demand, increasing the salt concentration near the soil surface. After the cropping season, the autumn and winter rainfall displace the salts from the topsoil towards the deeper soil layers and the water table from where it is exported by the drainage system. The differences observed in the vertical distributions of the soil parameters between 2017 and 2020 can be explained by the different antecedent rainfall conditions during both of the years (Figure 2). The heavy rainfall observed on days 18 and 19 before the survey of 2017 did a better job in leaching the $[Na^+]$ from the topsoil than the light rainfall between days 21 and 28 before the survey in 2020. Although the average $[Na^+]$ across the soil profile was the same for both of the surveys, $[Na^+]$ at 0.1, 0.3 and 0.5 m was 45, 56 and 39% higher, respectively, and 12 and 20% lower at 0.7 and 0.9 m, respectively, in 2020 as compared to 2017. Due to the small sample size and the large variability across the five sampling locations, these differences were only significant ($p < 0.05$) down to 0.5 m (Table 1). Although less prominent, the ECe was also higher in the deepest layer and lower at 0.3 and 0.5 m in 2017 (only significant at 0.3 m), indicating a better leached soil profile than in 2020. In addition, the $[Ca^{2+}]$ showed higher (but non-significant) concentrations in 2020 as compared to 2017. The pHe was higher in 2017 across the entire soil profile, although the differences were only significant at 0.5 and 0.7 m depth.

**Table 1.** Mean soil properties by soil depth and sampling point for soil samples taken in 2017 and 2020. Significant differences (ANOVA) are indicated by different letters ($p < 0.05$). Significance of the differences between 2017 and 2020 (Student's *t*-test) are shown in the lower panel. ECe and SAR are in dS m$^{-1}$ and mmol$_c$$^{1/2}$ L$^{-3/2}$, respectively, while ion concentrations are in mmol$_c$ L$^{-1}$.

| | | Mean | Depth (m) | | | | | Sampling Points | | | | |
|---|---|---|---|---|---|---|---|---|---|---|---|---|
| | | | 0.1 | 0.3 | 0.5 | 0.7 | 0.9 | 1 | 2 | 3 | 4 | 5 |
| 2017 | ECe | 3.62 | 1.62a | 1.61a | 2.50a | 4.92ab | 7.46b | 4.82a | 3.74a | 1.83a | 5.73a | 1.99a |
| | SAR | 24.0 | 8.3a | 10.5a | 14.7ab | 28.0ab | 58.6b | 42.7a | 19.7a | 11.1a | 31.8a | 14.9a |
| | Na$^+$ | 740 | 236a | 287a | 462a | 1045ab | 1662b | 975a | 706a | 373a | 1250a | 388a |
| | Ca$^{2+}$ | 38.4 | 37.9a | 23.1a | 44.3a | 48.1a | 38.8a | 37.7a | 38.6a | 38.2a | 45.6a | 32.0a |
| | Mg$^{2+}$ | 30.9 | 35.1a | 22.4a | 23.4a | 38.7a | 34.8a | 28.9a | 32.3a | 35.7a | 39.4a | 18.1a |
| | K$^+$ | 28.4 | 24.6a | 19.7a | 20.8a | 42.7a | 34.1a | 24.0a | 32.6a | 28.6a | 39.7a | 16.8a |
| | pHe | 8.6 | 8.4a | 8.5a | 8.9a | 8.7a | 8.4a | 9.0a | 8.7a | 8.3a | 8.4a | 8.5a |
| 2020 | ECe | 3.40 | 1.37a | 2.76ab | 2.89ab | 4.37ab | 5.60b | 2.88ab | 3.15ab | 1.62a | 5.92b | 3.43ab |
| | SAR | 17.6 | 8.8a | 15.5ab | 14.7ab | 22.7bc | 26.4c | 15.8a | 20.3a | 14.4a | 20.3a | 17.3a |
| | Na$^+$ | 735 | 343a | 448a | 644a | 919ab | 1323b | 642a | 848a | 444a | 1051a | 692a |
| | Ca$^{2+}$ | 57.2 | 60.0a | 46.2a | 69.4a | 47.5a | 62.8a | 58.0ab | 51.1ab | 31.7a | 80.4b | 64.8ab |
| | Mg$^{2+}$ | 45.5 | 35.7ab | 25.1a | 46.5ab | 49.9ab | 70.5b | 47.2a | 55.6a | 30.7a | 56.4a | 37.9a |
| | K$^+$ | 33.2 | 28.5a | 27.7a | 34.2a | 37.1a | 38.7a | 14.5a | 25.0ab | 44.0b | 41.3b | 41.5b |
| | pHe | 7.8 | 7.8a | 7.7a | 7.8a | 7.9a | 8.0a | 7.5a | 8.2b | 8.6bc | 7.4a | 7.4a |
| *t*-test | ECe | | 0.203 | 0.007 | 0.379 | 0.629 | 0.290 | 0.298 | 0.256 | 0.525 | 0.729 | 0.043 |
| | SAR | | 0.854 | 0.228 | 0.992 | 0.455 | 0.256 | 0.347 | 0.865 | 0.118 | 0.189 | 0.347 |
| | Na$^+$ | | 0.025 | 0.010 | 0.032 | 0.533 | 0.450 | 0.439 | 0.076 | 0.088 | 0.309 | 0.005 |
| | Ca$^{2+}$ | | 0.270 | 0.357 | 0.355 | 0.973 | 0.184 | 0.359 | 0.580 | 0.492 | 0.155 | 0.163 |
| | Mg$^{2+}$ | | 0.957 | 0.778 | 0.121 | 0.489 | 0.098 | 0.367 | 0.157 | 0.636 | 0.364 | 0.038 |
| | K$^+$ | | 0.709 | 0.291 | 0.130 | 0.678 | 0.490 | 0.194 | 0.413 | 0.148 | 0.742 | 0.004 |
| | pHe | | 0.237 | 0.056 | 0.028 | 0.029 | 0.389 | 2.0E-04 | 0.079 | 0.287 | 0.005 | 0.006 |

The comparison of the profile-averages of the soil properties between the different sampling points in 2017 yielded substantially higher (but non-significant) ECe and $[Na^+]$ at P4, P1 and P2 (in order of decreasing magnitude), as compared to P3 and P5. A similar

spatial pattern was observed in 2020, although in this case significant differences were only observed between P3 and P4 for ECe. In addition, the $[Ca^{2+}]$ showed a significantly higher concentration (150%) in P4 as compared to P3, while the pHe was significantly higher (pHe > 8) at P2 and P3 as compared to P1, P4 and P5 (pHe $\approx$ 7.4). The profile-averaged pHe was also higher in 2017 than in 2020, although only significantly at P1, P4 and P5. Only ECe, $[K^+]$, $[Na^+]$ and $[Mg^{2+}]$ were significantly higher in 2020 than in 2017 at P5.

### 3.2. Depth-Specific Increments of Soil Properties between 2017 and 2020

The effects of the less effective leaching in 2020 were also evident from the depth-specific increment patterns for ECe, $[Na^+]$ and SAR between 2017 and 2020 for the five sampling points (Figure 3). The ΔECe profiles (Figure 3a) showed overall positive increments for the top 0.5 m, indicating a higher ECe in 2020, while a tendency towards negative ΔECe was observed for the deepest soil layers. The point with the largest ECe for both of the surveys (P4) showed a maximum ΔECe at 0.5 m, while the maximum for the remaining points occurred at 0.3 m. P5, which was located farthest from the tile-drain outlets, showed positive ΔECe across the entire soil profile. A similar trend with depth was observed for SAR (Figure 3b) and $[Na^+]$ (Figure 3c), evidencing the dominant effect of $[Na^+]$ on ECe and SAR. P1 showed the largest reduction in ECe, $[Na^+]$ and SAR at 0.7 and 0.9 m. This point is located near the tile-drain outlets near the northern limit of the field where the altitude was lowest and might have accumulated the $[Na^+]$ that was leached from the topsoil in 2017. During the subsequent seasons, this $[Na^+]$ was further leached and exported through the drainage system, resulting in substantially smaller values of ECe, $[Na^+]$ and SAR in 2020.

The pHe profiles (Figure 3g) showed a general decrease in pHe from 2017 to 2020 at all of the depths, with the largest drop between 0.3 and 0.7 m, except for the point with the smallest ECe and $[Na^+]$ (P3), where the pHe increased at 0.1 and 0.9 m and remained the same at the intermediate depths. The $\Delta K^+$, $\Delta Mg^{2+}$ and $\Delta Ca^{2+}$ profiles showed different patterns as compared to the salinity-related soil properties. P5, located farthest away from the tile-drainage outlets, showed positive increments across the entire soil profile for all of the cations. The $\Delta K^+$ increased between 2017 and 2020 across most of the soil profiles in P5 and P3, the points with the smallest ECe and $[Na^+]$. For the $\Delta Mg^{2+}$ and $\Delta Ca^{2+}$ no clear common patterns with depth or across the different sampling points could be devised, except for P2 and P4 below 0.3 m. The point- and depth-specific differences in the mobility of these cations, possibly modulated by pHe variations, might explain this behavior.

### 3.3. PCA of Soil Properties in 2017 and 2020

For 2017 and 2020, the first and second PC explained 46.2 and 29.7% and 52.4 and 20.4% of the total variance, respectively (Figure 4). For both of the years, the PC1 appeared to be related to soil salinity, while PC2 was linked to pHe. The strongest correlations between ECe and $[Na^+]$ (R=0.99) and $[Ca^{2+}]$ and $[Mg^{2+}]$ (R = 0.65) were found in 2017, possibly because of the stronger leaching of the soil profile due to the rainfall before this survey (Figure 2). A clear antagonism was also observed in 2017 between the pHe and the cations $[Ca^{2+}]$, $[Mg^{2+}]$, and $[K^+]$, with this relationship being less evident in 2020. For both of the years, most of the soil samples taken from 0.1 to 0.5 m depth grouped along the positive side of the PC1 axis, in accordance with the less saline conditions as compared to the deeper soil layers ($\geq$0.7 m). In 2020 (Figure 4b), all of the considered variables, except $[Ca^{2+}]$ and pHe, projected closely to the PC1 axis and showed stronger correlations, as compared to 2017, while the antagonism between $[Ca^{2+}]$ and pHe along PC2 was conserved. This indicates that the variations in ECe in 2017 were mainly attributable to the variations in $[Na^+]$, while in 2020 $[Mg^{2+}]$ and $[K^+]$ also contributed to the variation in ECe.

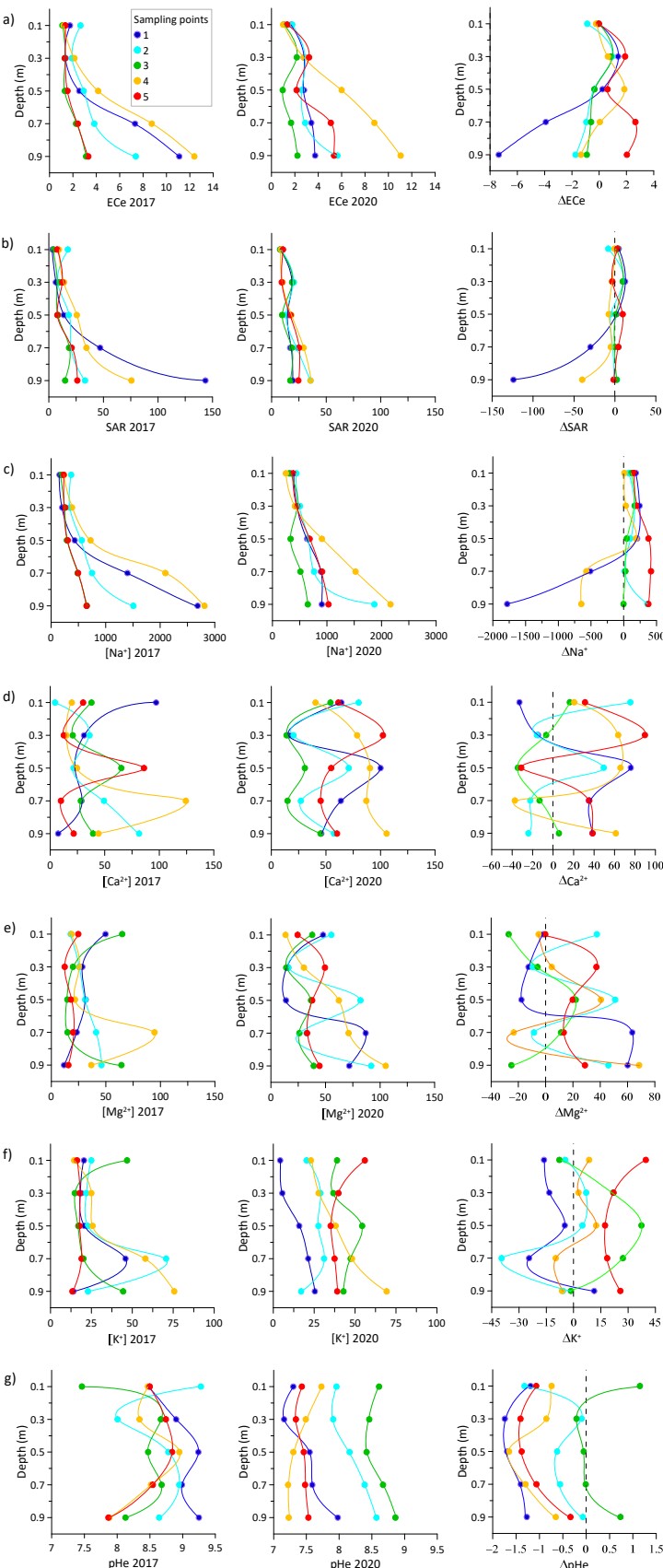

**Figure 3.** Distributions of (**a**) ECe; (**b**) SAR; (**c**) [Na$^+$]; (**d**) [Ca$^{2+}$]; (**e**) [Mg$^{2+}$]; (**f**) [K$^+$]; and (**g**) pHe and increments (Δ) between 2017 and 2020, across the soil profile at the five sampling points. ECe is in dS m$^{-1}$; concentration of cations in mmol$_c$ L$^{-1}$.

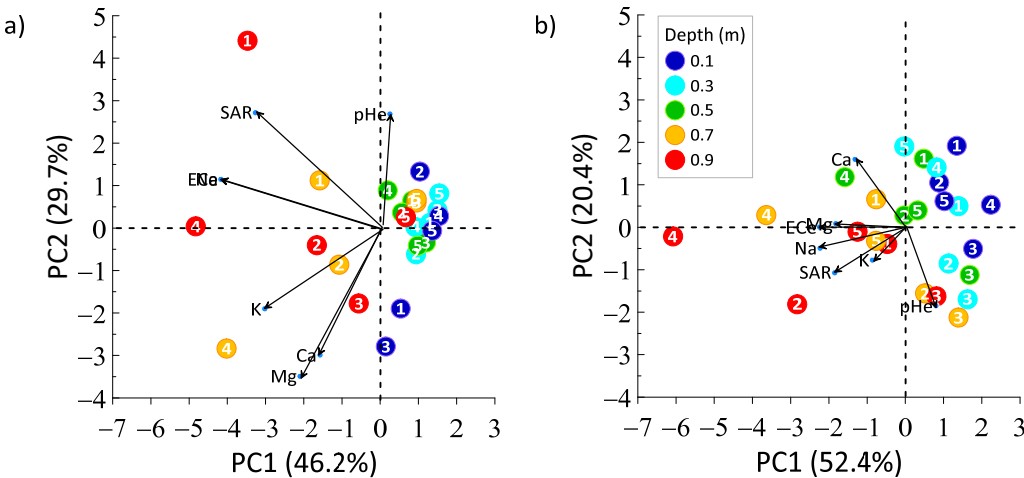

**Figure 4.** Biplot with results of the PCA for the considered soil properties (pHe, ECe, SAR, [Na$^+$], [Ca$^{2+}$], [Mg$^{2+}$] and [K$^+$]) in (**a**) 2017 and (**b**) 2020. Individual soil samples are represented by point numbers (1–5, Figure 1) and depth.

Overall, the differences in the association between the considered soil properties between 2017 and 2020 were a result of the different antecedent rainfall conditions, leading to waterlogging and intense [Na$^+$] leaching of the soil profile in 2017, as compared to less intense rainfall and leaching in 2020. The contrasting behavior of the different sampling points (e.g., P4) can be attributed to differences in the soil matrix composition and structure, originating from the original terrain characteristics of the marshes that consisted of a network of lower, often water-logged, areas (*lucios*) in-between the elevated areas where the reduction conditions were less frequently met.

### 3.4. ECa in 2017 and 2020

The average ECa-1 and ECa-3 signals were 26 and 17% higher in 2017 than in 2020, respectively, indicating more conductive conditions overall in 2017 (Table 2). In 2017 and 2020, the ECa-1/ECa-3 ratios were 0.84 and 0.78, respectively, indicating a slightly less homogeneous conductivity across the soil profile in 2020. The spatial variability (CV) was higher in 2020 for both of the signals, which points towards a more homogeneous conductivity across the field in 2017. For both of the surveys, the CV was smaller for the deep signal (ECa-3) than for the shallow signal (ECa-1), indicating a more homogenous conductivity distribution in the deeper soil layers than in the shallower ones.

**Table 2.** Descriptive statistics of interpolated ECa-1 and ECa-3 (mS m$^{-1}$) in 2017 and 2020.

|  | ECa-1 | | ECa-3 | |
| --- | --- | --- | --- | --- |
|  | **2017** | **2020** | **2017** | **2020** |
| m* | 474.0 | 376.0 | 566.8 | 484.2 |
| min | 204.9 | 96.1 | 337.1 | 266.3 |
| max | 831.0 | 753.1 | 790.0 | 780.7 |
| med | 477.5 | 376.3 | 574.4 | 491.1 |
| s | 118.7 | 109.3 | 98.3 | 100.0 |
| CV | 0.25 | 0.29 | 0.17 | 0.21 |
| kurtosis | −0.826 | −0.779 | −0.885 | −0.866 |
| skewness | 0.019 | 0.128 | −0.185 | −0.121 |

m*: mean; med: median; s: standard deviation; CV: coefficient of variation.

The changes in the ECa between 2017 and 2020 were evaluated for the top 1 m, approximately down to the drainage tiles, using the ECa-1 signal. Figure 5a,b shows the spatial distribution of the ECa-1 measured in 2017 and 2020, respectively. Both of the maps show similar ECa patterns, with a high ECa near the adjacent road which impedes lateral

soil water flow. In addition, towards the SW limit of the field, along part of the irrigation service road, large ECa values were observed. The ECa was generally smaller towards the northern limit, where the tile outlets discharge in the drainage ditch. The transect with the five sampling points crossed an area of low ECa, intersected by the fringes of high ECa. P3 and P5 were in the former, while P4 was in the latter. Note that, in accordance with the large ECa, P4 showed the highest ECe and cation concentrations for both of the years (Table 1). Figure 5c shows the increment of ECa-1 from 2017 to 2020 (ΔECa). The smallest increments (near zero) were observed in the areas with the smallest ECa values. The histogram and cumulative distribution of ΔECa-1 m (Figure 5d) shows that ECa-1 decreased from 2017 to 2020 in most of the field. About 50% of the field area showed $\Delta ECa-1 < -100$ mS m$^{-1}$.

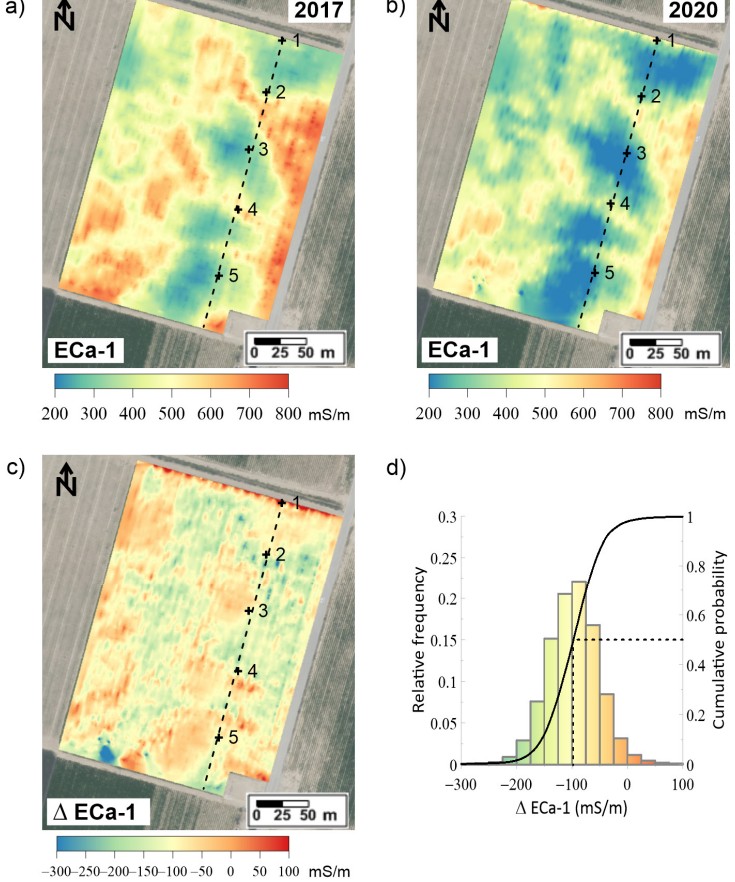

**Figure 5.** Spatial distribution of ECa-1 in (**a**) 2017 and (**b**) 2020; (**c**) Increments of ECa-1 from 2017 to 2020 (ΔECa-1); and (**d**) histogram and cumulative distribution of ΔECa-1.

### 3.5. 2D inversion of EMI Datasets: EC Variations along a Transect and Calibration

Figure 6a,b shows the EC images of the soil profile along the transect and EC curves for the five sampling points (Figure 1) for 2017 and 2020, respectively. The largest differences between both surveys were observed in the top 0.5 m of the soil profile, with a deeper penetration of zones with a small EC in 2020. Yet, the strongest absolute increments occurred below 0.5 m. (Figure 6c). P3 and P5 were in areas of a small EC, unlike P4 and P2. This contrast was also observed in the analysis of ECe and [Na$^+$] from P3 and P5 with respect to P4 (Table 1). Overall, the EC decreased from 2017 to 2020 across the entire soil profile, except for P1 where ΔEC was positive for all of the depths (Figure 6c). This contrasts with the observed increments for ECe in Figure 3b. Despite similar depth-specific ECe and EC patterns in 2017 (see also Table 1), P1 showed the largest EC in 2020 (Figure 6b), while ECe was largest in P4 (Table 1). The wetter soil conditions during the ECa survey in 2017 resulted in better EC–ECe relationships, as a higher proportion of the present ions entered in dissolution in the wet soil, approximating better saturated paste conditions under which

the ECe is measured, as compared to the drier soil in 2020. In addition, the position of P1 near the tile-drain outlets and the drainage ditch might have led to locally wetter soil near P1 in 2020, resulting in higher EC as compared to the other four points.

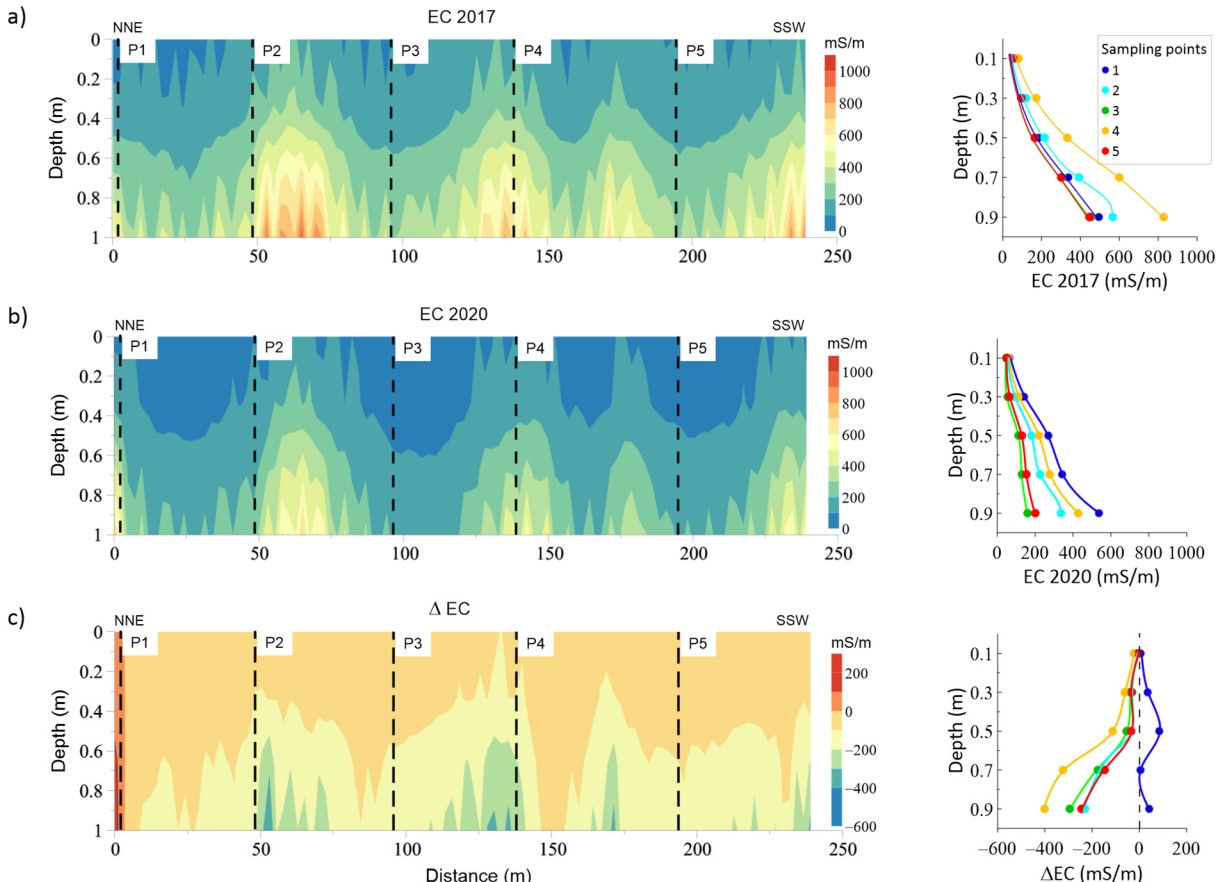

**Figure 6.** Images of inverted ECa (EC) along the transect for (**a**) 2017; (**b**) 2020; and (**c**) increments of EC between 2017 and 2020 (ΔEC).

Strong correlations were observed between the EC and ECe, $[Na^+]$ and SAR for 2017 and 2020 (Table 3), unlike $[Ca^{2+}]$, $[Mg^{2+}]$ and $[K^+]$ for which correlations with the EC were smaller and less consistent between both years. This indicates that the EC showed a strong potential for estimating depth-specific ECe, $[Na^+]$ and SAR. Yet, the shallowest signal provided by the used EMI sensor has a theoretical depth of exploration near 0.5 m, resulting in less detailed and reliable estimates of the EC near the soil surface (0.1 m depth). To check this further, the depth-specific Pearson correlation coefficient was calculated, as shown in Table 3. The strongest correlations of ECe and SAR with EC were found for the deeper soil layers, and for the wettest year (2017). In the topsoil (0.1 and 0.3 m) no correlation was found between EC and ECe or SAR, except at 0.3 m in 2017. In the wetter soil, in 2017, the cations were possibly better dissolved, leading to a more conductive soil solution as compared to 2020.

The relationships between the EC, ECe and SAR for 2017 and 2020 (Figure 7) showed that, although the EC values were higher in 2017 than in 2020, particularly in the deepest soil layers (Figure 7a), the ECe and SAR showed a different tendency, with higher ECe and SAR at intermediate depths in 2020 at certain sampling points. These apparently different relationships for the EC with respect to the ECe and SAR might be the result of differences in the water content across the soil profile.

**Table 3.** Pearson correlation coefficient (R) between depth-specific EC and soil chemical properties measured in 2017 and 2020, and correlations by depths.

| | | Depth (m) | | | | | |
|---|---|---|---|---|---|---|---|
| | | 0–0.9 | 0.1 | 0.3 | 0.5 | 0.7 | 0.9 |
| 2017 | ECe | 0.91 | 0.05 | 0.91 | 0.96 | 0.99 | 0.98 |
| | SAR | 0.88 | 0.33 | 0.72 | 0.96 | 0.93 | 0.97 |
| | $Na^+$ | 0.91 | 0.22 | 0.94 | 0.94 | 0.98 | 0.99 |
| | $Ca^{2+}$ | 0.46 | −0.51 | −0.09 | −0.74 | 0.98 | 0.24 |
| | $Mg^{2+}$ | 0.46 | −0.58 | 0.71 | 0.33 | 0.99 | −0.06 |
| | $K^+$ | 0.63 | −0.55 | 0.91 | 0.93 | 0.66 | 0.82 |
| | pHe | −0.19 | 0.35 | −0.50 | 0.67 | −0.25 | −0.17 |
| 2020 | ECe | 0.89 | 0.12 | 0.27 | 0.92 | 0.73 | 0.91 |
| | SAR | 0.88 | 0.04 | −0.22 | 0.55 | 0.81 | 0.94 |
| | $Na^+$ | 0.95 | −0.18 | 0.06 | 0.86 | 0.82 | 0.99 |
| | $Ca^{2+}$ | 0.36 | −0.14 | 0.19 | 0.97 | 0.76 | 0.85 |
| | $Mg^{2+}$ | 0.73 | −0.22 | −0.05 | 0.75 | 0.84 | 0.99 |
| | $K^+$ | 0.35 | −0.74 | −0.91 | −0.55 | 0.00 | 0.35 |
| | pHe | −0.18 | −0.45 | −0.45 | −0.53 | −0.52 | −0.51 |

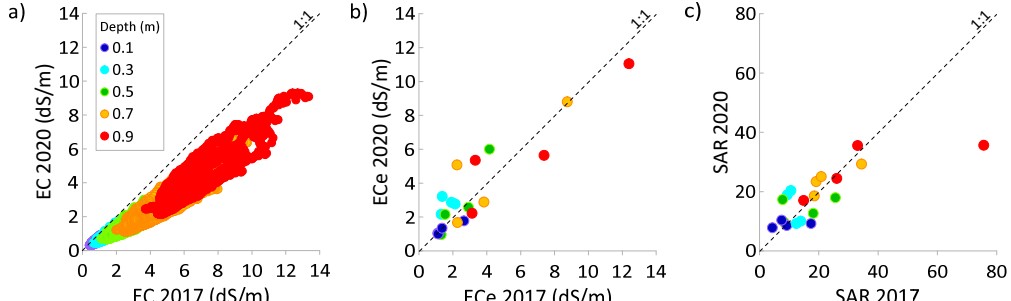

**Figure 7.** Relationships between (**a**) EC; (**b**) ECe; and (**c**) SAR for 2017 and 2020.

Due to the anomalous behavior of P1 (Figure 6c), this sampling point was excluded for calibration purposes. This point is located at the headland near the service track and the drainage ditch (Figure 1), where during winter, a surface drainage pipe is provisionally installed to evacuate the surface water from the field and alleviate the ponding, causing contrasting hydrological conditions with respect to the remainder of the field. The linear calibration equations for ECe against EC (Figure 8a,b) were closer to the 1:1 line in 2017 than in 2020, when a steeper slope was obtained. The calibration equations near the 1:1 line can be associated with field measurement conditions similar to those for which the saturated paste extract was obtained in the laboratory. Under such conditions, a similar proportion of the ions enter in dissolution in the field soil water as in the laboratory saturated paste, leading in theory to similar ECe and EC. As the soil dries, the ions precipitate and their concentration in the soil solution decreases and so does the field-measured EC, resulting in a steeper slope for the 2020 ECa–EC calibration equation as compared to 2017. This also supports the hypothesis of wetter soil conditions and a more conductive soil in 2017 as compared to 2020 (see also Figure 2). The largest $R^2$ was obtained for the linear equations, ranging from 0.77 to 0.83 for ECe and SAR (Figure 8c,d).

The validation (LOOCV) of the regression models yielded better results for the ECe than the SAR (Table 4). The NSE indicated overall "acceptable" results for ECe in both of the years, while the results for SAR were only "acceptable" in 2020. The same results were obtained when considering only the subsoil (0.5–0.9 m layers), while "unsatisfactory" validation results were obtained for the topsoil (0.1 and 0.3 m layers) in all of the cases. The overall better performance obtained for ECe is possibly a result of the strong relationship between ECe and $Na^+$, and the common pattern of both variables with depth (Table 1 and Figure 4), according to the increasing EC with depth (Figure 6). In contrast, the SAR

also depends on the other cations ($Ca^{2+}$, $Mg^{2+}$) that lacked a clear pattern with depth and therefore showed weaker correlations with the EC. Therefore, it might be beneficial to estimate first the $Na^+$, $Ca^{2+}$, $Mg^{2+}$ from the EC and then calculate the SAR.

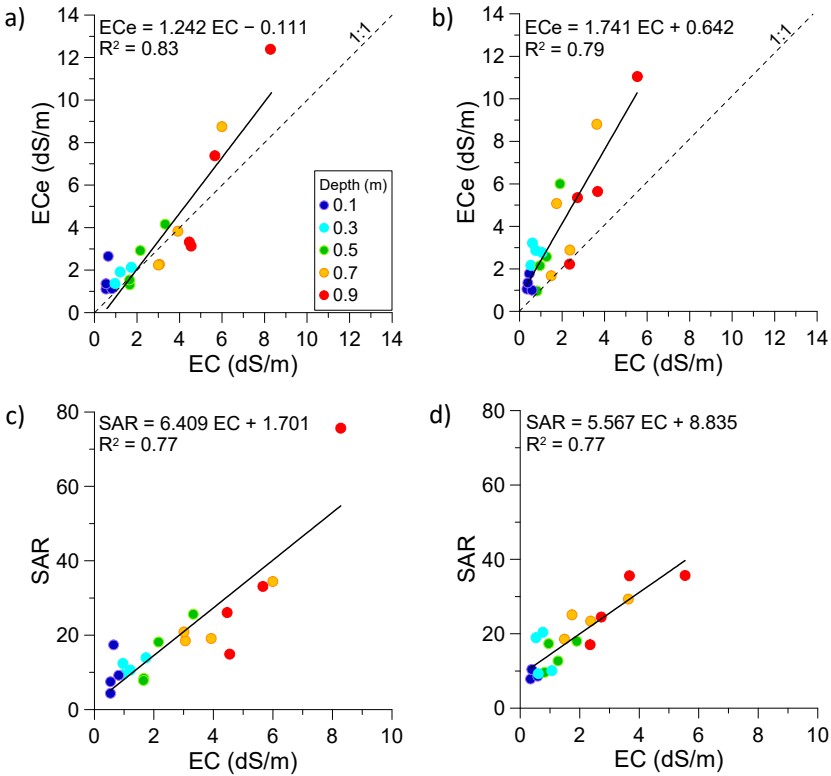

**Figure 8.** Relationships between (**a**) EC and saturated paste extract EC (ECe) for 2017 and (**b**) 2020 and between (**c**) EC and sodium absorption ratio (SAR) for 2017 and (**d**) 2020.

**Table 4.** Leave-one-out cross validation parameters for the estimation of ECe and SAR in 2017 and 2020, distinguishing between top- (0.1 and 0.3 m layers) and subsoil (0.5–0.9 m layers).

| EC | ECe | | | SAR | | |
|---|---|---|---|---|---|---|
| | RMSE | MAE | NSE | RMSE | MAE | NSE |
| 2017 | 1.44 | 1.08 | 0.74 | 14.67 | 9.16 | 0.57 |
| Topsoil | 0.61 | 0.47 | −2.05 | 4.33 | 3.03 | −1.30 |
| Subsoil | 1.4 | 1.23 | 0.71 | 11.6 | 8.26 | 0.51 |
| 2020 | 1.35 | 1.09 | 0.74 | 6.85 | 5.75 | 0.70 |
| Topsoil | 0.86 | 0.7 | 0.06 | 7.16 | 5.75 | −0.12 |
| Subsoil | 1.59 | 1.35 | 0.69 | 4.78 | 3.89 | 0.67 |

Despite the limited availability of analytical soil data, these validation results illustrate the potential of EMI tomography for soil salinity monitoring in hyper-conductive environments, such as the B-XII irrigation district. The poor performance in the topsoil can be partly attributed to the small lateral variation of the shallow soil properties (small range of data values), as compared to the vertical variation across the soil profile. Further research should involve EMI instruments that provide more detail near the soil surface, which can be achieved by multi-frequency or multi-coil instruments with smaller theoretical depths of exploration.

The depth-specific ECe and SAR were estimated for 2017 and 2020 using the 3D inversion results and the relationships shown in Figure 8. (Figure 9). Considering the entire field, the ECe increased in 2020 as compared to 2017, while the SAR overall showed an opposite trend. The largest changes occurred along the eastern limit of the field, along the

road which constrained the lateral water and solute flow out of this area. In the SW corner of the field, a higher ECe was also estimated in 2020. As expected, the ECe increased with depth for both surveys. As a result of the differences in the antecedent leaching conditions between the surveys of 2017 and 2020, the SAR was substantially lower in 2020, at a 0.7 and 0.9 m depth (Figure 9d,e), while similar values were obtained in both of the years at 0.5 m (Figure 9d). These patterns and variations can be explained in terms of the [Na$^+$] leaching dynamics across the field between 2017 and 2020 (see also Table 1). The maps in Figure 9 allow a detailed assessment of the evolution of the depth-specific salinity status across the field and are relevant for informing site-specific soil and water management decisions during the growing season.

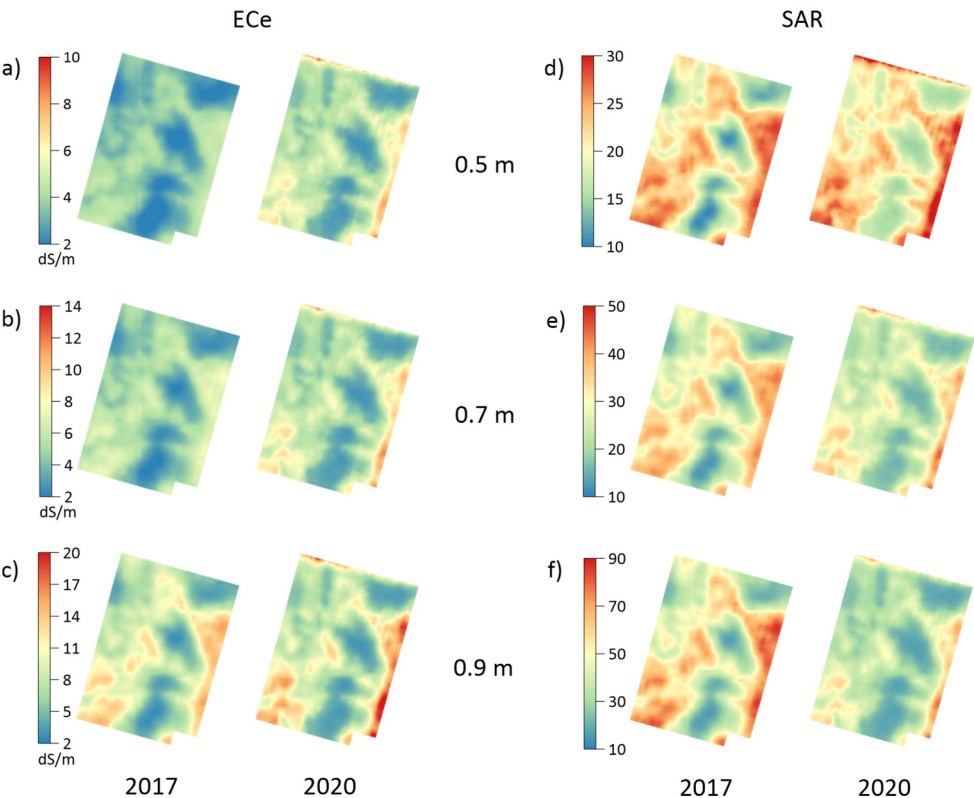

**Figure 9.** ECe (**a–c**), and SAR (**d–f**) for 2017 and 2020 at depths of 0.5, 0.7 and 0.9 m.

*3.6. Correlation between Depth-Specific EC and NDVI*

The correlations between the four ECa signals and NDVI were similar for both survey years. The correlations between the depth-specific EC and NDVI were also similar to those obtained for ECa. In tomato, on the day of the maximum NDVI, the correlations with the four ECa signals from 2017 ranged from −0.65 to −0.63, while the correlations with the depth-specific EC ranged from −0.66 to −0.64. In cotton, these correlations ranged from −0.83 to −0.81 and from −0.86 to −0.77, and in sugar beet from −0.69 to −0.66 and from −0.73 to −0.64, respectively. In general, the correlation between the depth-specific EC and NDVI increased slightly with the soil depth.

The negative correlations between the NDVI and depth-specific EC were obtained throughout the different cropping seasons for both of the survey years (Figure 10). The correlations were only strong (R <−0.5) during the second half of the tomato-growing period (2017), during the entire cotton-growing period (2018) and occasionally in the spring during the sugar beet-growing period (2019). The negative correlations indicate that the crop development, as measured by NDVI, was limited where the EC was largest, particularly during the warmest periods of the cropping season, when the evapotranspiration and irrigation rates were highest. This supports the hypothesis that the spatial salinity pattern in the topsoil, which is expected to affect the crop development and control the spatial

NDVI pattern, originates from the deeper soil layers, because of the capillary effects during the periods of strong atmospheric water demand. The weaker correlations observed for sugar beet could be a consequence of the different growing season as compared to tomato and cotton. The sugar beet was grown in spring and harvested in early July, with irrigation being cut-off several weeks before the harvest. As a result, the environmental conditions were less favorable for the capillary rise of saline soil water towards the topsoil than in the other crops.

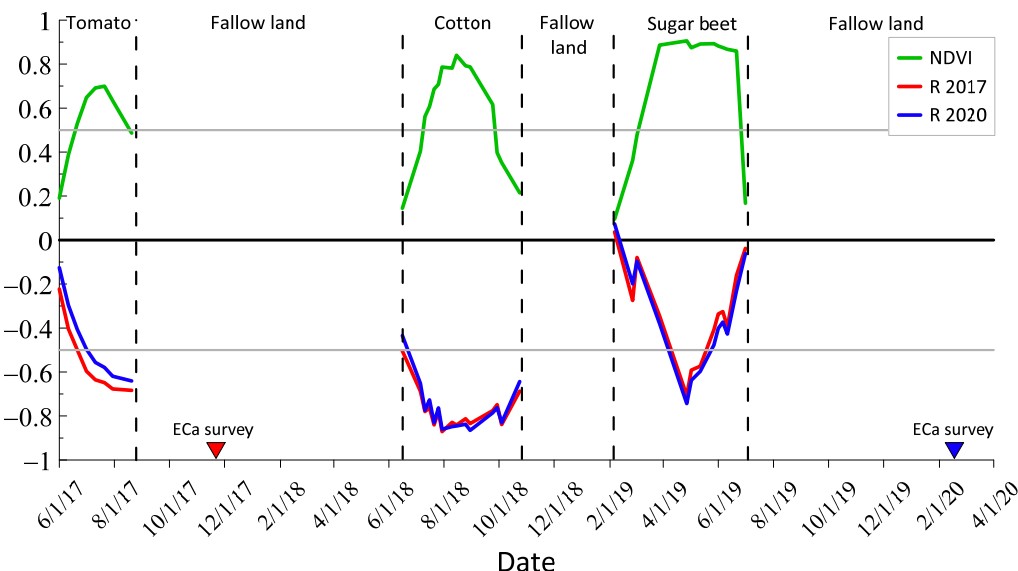

**Figure 10.** Evolution of mean NDVI and Pearson correlation coefficient (R) between NDVI and depth-specific EC at 0.9 m depth, measured in 2017 and 2020.

Figure 11 shows the NDVI images on the day of maximum NDVI and the corresponding inverse relationships between NDVI and EC at 0.9 m for the three crops. The cotton and sugar beet showed a concave type of relationship ($R^2$ = 0.75 and 0.53, respectively), while a convex power law type of relationship could be fitted for processing tomato, despite the large data dispersion ($R^2$ = 0.50), which indicates the existence of other effects on the NDVI besides the EC. The NDVI ranged from 0.55 to 0.85 for EC < 700 mS m$^{-1}$, while the NDVI was limited to 0.75 in areas with EC > 700 mS m$^{-1}$. In cotton and sugar beet, the negative effects of the EC on the NDVI were mainly appreciable in areas with EC > 700 mS m$^{-1}$ (Figure 11b,c).

Table 5 shows the correlations between the depth-specific EC and NDVI, measured along the transect, for each crop from seedling emergence/transplanting until reaching the maximum NDVI. The negative correlations became progressively stronger as the growing season evolved, possibly because of the plant roots growing deeper into more saline soil horizons. This evolution was also observed for sugar beet, despite the limited availability of NDVI imagery in the spring of 2019, due to cloudy weather conditions.

The strongest correlations with depth-specific EC ranged from −0.90 to −0.85 in cotton, from −0.73 to −0.71 in processing tomato and from −0.57 to −0.53 in sugar beet, corresponding roughly with the findings reported earlier on for the entire field.

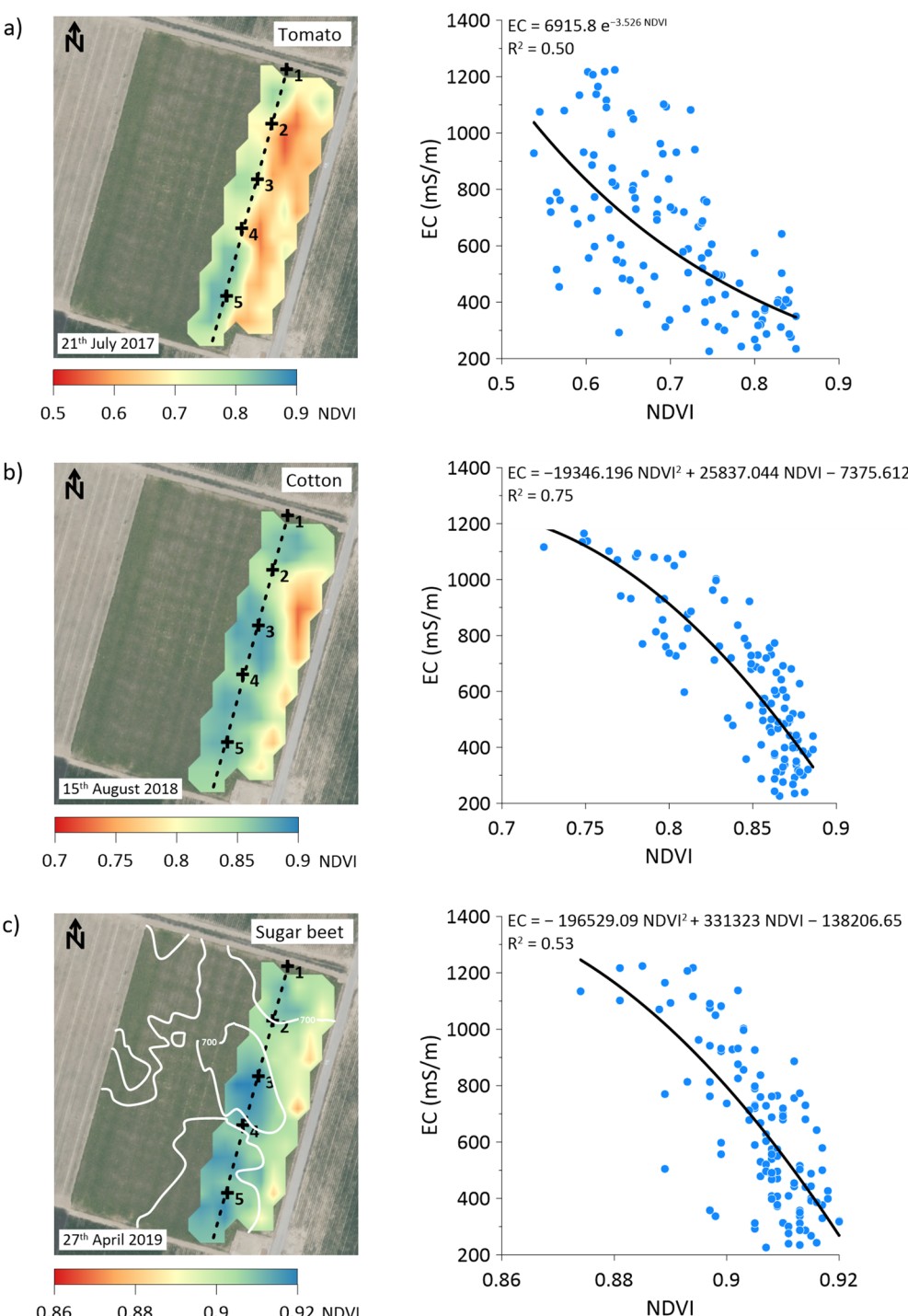

**Figure 11.** NDVI patterns and NDVI–EC relationships with depth-specific EC at 0.9 m measured in 2017 on the day of maximum NDVI for (**a**) processing tomato; (**b**) cotton; and (**c**) sugar beet. The white line represents EC= 700 mS m$^{-1}$ at 0.9 m depth.

The NDVI was generally smaller (indicating that plant development lagged behind) along the sections of the transect with the largest ECe, as compared to those with the smallest ECe at 0.9 m depth (Figure 12). As compared to cotton and sugar beet, processing tomato showed the strongest dependence on ECe from day 25 on (Figure 12a). The cotton showed a strong dependence on ECe throughout the entire period, even in the early crop growth stages (e.g., days 10–20), as also shown by the evolution of R (Figure 12b). The sugar beet developed more homogeneously along the transect, as compared to tomato and

cotton, and appeared to be less dependent on ECe (Figure 12c), which is also apparent from the weak R. Apart from the differences in the physiological response of these crops to soil salinity [34], a plausible explanation for these results is that sugar beet is grown during spring and harvested in early summer, while tomato and cotton are grown throughout the summer, when the largest amounts of water are applied to satisfy the maximum crop water requirements. The intense and persistent irrigation during this period decreases the depth of the saline water table, allowing brackish water to reach the root zone from where the water is transpired and evaporated, leaving behind the salts it contained. The conductivity hotspots detected for both years in Figure 6 represent the zones with salinization-prone topsoil due to differences in the soil water transport and retention characteristics, as compared to the surrounding areas of lower EC. As a result, the homogeneous irrigation along the transect might cause a local lifting of the water table or promote the occurrence of local perched water tables, which enhance the risk of topsoil salinization with possible negative effects for crop development [35].

**Table 5.** Pearson correlation coefficient (R) between NDVI and depth-specific EC, measured along the transect, from emergence/transplanting until reaching the maximum NDVI.

| R | | **2017** | | | | |
|---|---|---|---|---|---|---|
| | **Days since Emergence/Transplanting** | **EC 0.1** | **EC 0.3** | **EC 0.5** | **EC 0.7** | **EC 0.9** |
| Tomato 2017 | 10 | −0.14 | −0.16 | −0.15 | −0.16 | −0.17 |
| | 20 | −0.38 | −0.39 | −0.38 | −0.38 | −0.40 |
| | 30 | −0.54 | −0.55 | −0.55 | −0.54 | −0.56 |
| | 40 | −0.60 | −0.61 | −0.60 | −0.60 | −0.61 |
| | 50 | −0.66 | −0.68 | −0.67 | −0.67 | −0.68 |
| | 60 | −0.71 | −0.73 | −0.72 | −0.72 | −0.73 |
| Cotton 2018 | 5 | −0.39 | −0.40 | −0.41 | −0.42 | −0.43 |
| | 20 | −0.47 | −0.49 | −0.49 | −0.50 | −0.51 |
| | 25 | −0.67 | −0.69 | −0.69 | −0.70 | −0.71 |
| | 30 | −0.59 | −0.62 | −0.62 | −0.62 | −0.65 |
| | 35 | −0.70 | −0.73 | −0.73 | −0.74 | −0.76 |
| | 40 | −0.79 | −0.79 | −0.79 | −0.79 | −0.80 |
| | 45 | −0.79 | −0.84 | −0.84 | −0.85 | −0.87 |
| | 55 | −0.53 | −0.59 | −0.59 | −0.60 | −0.63 |
| | 60 | −0.85 | −0.88 | −0.88 | −0.88 | −0.90 |
| Sugar beet 2019 | 5 | 0.11 | 0.12 | 0.11 | 0.11 | 0.11 |
| | 30 | 0.07 | 0.06 | 0.06 | 0.05 | 0.03 |
| | 60 | −0.35 | −0.42 | −0.41 | −0.43 | −0.46 |
| | 70 | −0.57 | −0.55 | −0.55 | −0.55 | −0.53 |

The observed link between NDVI and EC opens an avenue for future research on salinity assessment in this hypersaline environment. According to our findings, the EMI-inferred EC patterns appear to be time stable and have to be measured only once. The correlation of EC with the freely available NDVI imagery can then be calculated throughout preceding or subsequent growing seasons and can be used as an indicator to compare, monitor and assess the evolution of soil salinity and its effects on crop development throughout or across different growing seasons and crops. In addition, the availability of yield monitoring data could further unlock the potential of this methodology and allow the EC–NDVI correlation to be ultimately related to crop productivity.

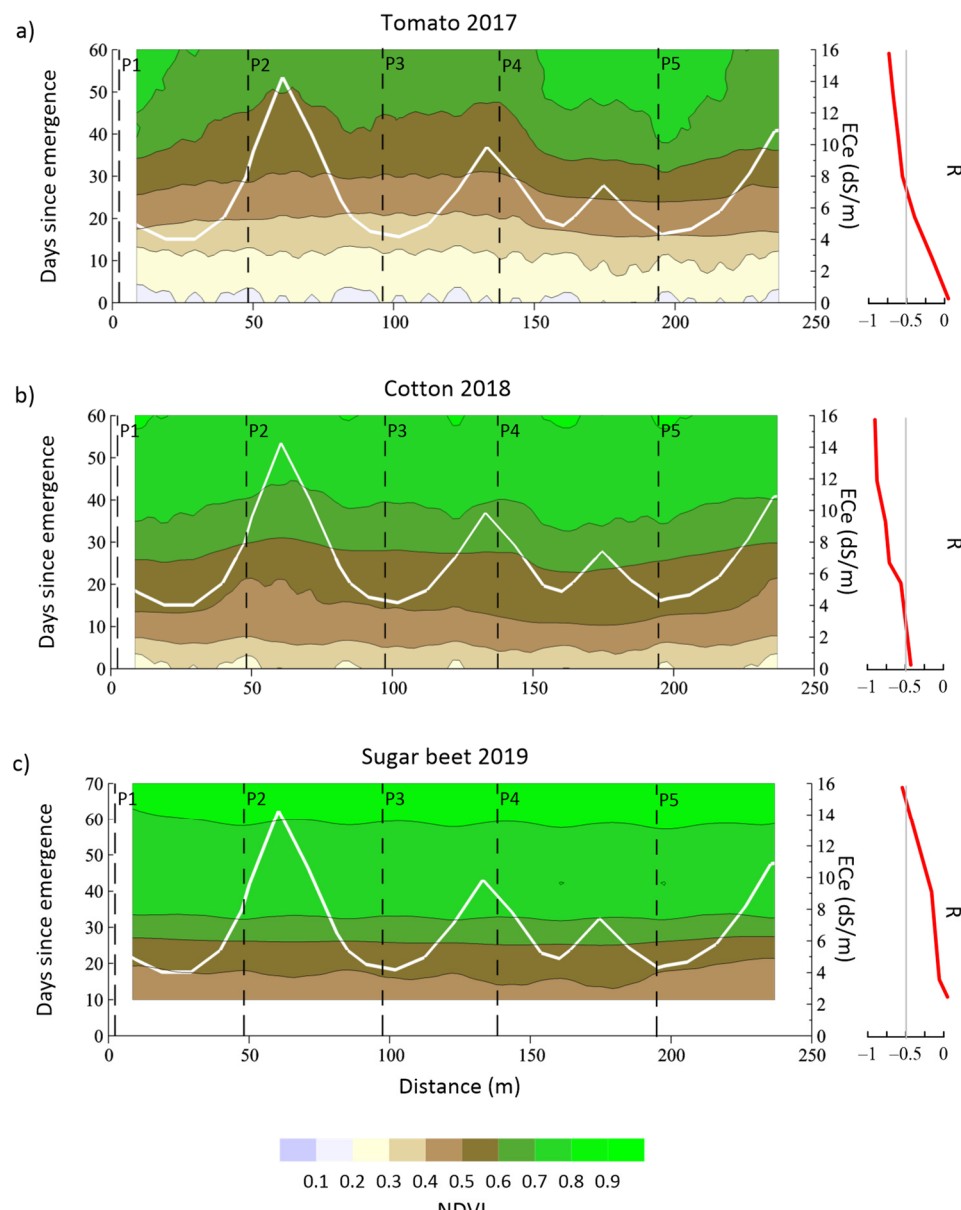

**Figure 12.** Evolution of NDVI along the transect for (**a**) processing tomato; (**b**) cotton; and (**c**) sugar beet from the day of seedling emergence/transplanting to the day with maximum NDVI. The white and red lines represent estimated ECe at 0.9 m depth in 2017 and its correlation (R) with NDVI, respectively.

## 4. Conclusions

The EMI surveys performed in 2017 and 2020 in the study field showed similar spatial ECa patterns, with the largest average ECa in 2017, which was attributed to wetter soil conditions in 2017, as a result of differing antecedent weather conditions. The temporal persistence of the spatial ECa patterns indicates that the implemented soil and water management when cropping cotton and sugar beet did not affect the spatial ECa distribution; therefore, this is attributable to the underlying soil variability resulting from the original marsh terrain characteristics before land reclamation or the presence of infrastructures that limit lateral water and solute movement near the limits of the field (e.g., roads). Partial clogging of tile drains, particularly in the SW part of the field, might also explain the presence of areas with high ECa.

The soil chemical analysis in 2017 and 2020 showed different vertical distributions of $Na^+$-related salinity indicators ($[Na^+]$, ECe and SAR) in the topsoil (<0.4 m) in both years,

with a better leached topsoil in 2017. The profile-averaged salinity indicators reproduced the observed ECa pattern at the five sampling points, indicating that EMI is successful in estimating the spatial salinity patterns in this hyper-conductive soil. In accordance with this, the depth-specific EC estimates obtained by inversion of the ECa measurements showed the strongest correlations with the salinity indicators for the deepest soil layers (>0.3 m), particularly for the wettest year (2017), while for the topsoil no clear relationships could be devised, possibly as a result of the small data ranges of the topsoil properties and the relatively large depth of exploration of the shallowest available EMI signal (~0.5 m), providing only limited vertical resolution in the EC estimates near the soil surface. In spite of these limitations, calibration relationships could be built that allowed the estimation of ECe and SAR from the inverted EC images, with $R^2$ near 0.80 and 0.77, respectively. EMI instruments that provide more detail near the soil surface and under wet soil conditions are expected to improve the performance of this method, allowing more accurate salinity estimation near the soil surface.

Although the estimated salinity levels in the rootzone were moderate and presumably harmless for crop development, an inverse relationship was observed between EC and Sentinel 2 NDVI imagery obtained during the growing seasons of processing tomato, cotton and sugar beet, indicating that the underlying ECa and depth-specific EC patterns affected crop development progressively, as the growing season evolved. The strongest relationships were found on the days when the average NDVI reached a maximum. This opens an avenue for future research by using the inverse relationship between the depth-specific EC and NDVI as an indicator for mapping and monitoring soil salinity effects on crop development across fields, growing seasons and crops, in environments such as the B-XII irrigation district.

**Author Contributions:** Data curation, A.G.J., M.R.R. and J.L.G.F.; Investigation, J.L.G.F., J.F.H.G., P.C.S. and K.V.; Methodology, M.R.R., M.F., J.F.H.G., B.S.B. and K.V.; Project administration, K.V.; Resources, K.V., J.F.H.G. and P.C.S.; Software, A.G.J., J.L.G.F. and M.F.; Supervision, K.V., M.F. and B.S.B.; Writing—original draft, J.L.G.F. and K.V. All authors have read and agreed to the published version of the manuscript.

**Funding:** This work is funded by the Spanish State Agency for Research through grants PID2019-104136RRC21, PID2019-104136RR-C22 and PhD grant PRE2020-095133 and by IFAPA/FEDER through grant AVA2019.018.

**Institutional Review Board Statement:** Not applicable.

**Informed Consent Statement:** Not applicable.

**Data Availability Statement:** The data are available from the authors upon request.

**Conflicts of Interest:** The authors declare no conflict of interest.

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
