# Peer review of "Depth-Specific Soil Electrical Conductivity and NDVI Elucidate Salinity Effects on Crop Development in Reclaimed Marsh Soils"

_remotesensing, doi:10.3390/rs14143389_

Round 1

Reviewer 1 Report

In this study, the authors proposed a framework for evaluating the field-scale soil salinity based on EMI and remote sensing. There are some issues that need to be resolved before this manuscript can be considered for publication. My recommendation is MAJOR revision.
General comments:

  1. As REOTE SENSING is an international journal, it seems to be better to give soil names in accordance with international soil classification (WRB).
  2. The language needs to be edited by native speakers. The use of language should be carefully checked. There are many unnecessary descriptions. There are more deficiencies in language use. English polishing is must.
  3. In general, the logics among all sections are poor. Therefore, I can hardly get your objectives in this review. However, it is missing two important elements, i.e., (i) Hypothesis explanation, and (ii) Novelty statement. These two important mentions will guide the reader towards the heart of the article.
  4. The abstract lacks the necessary structure, usually the abstract includes background, purpose, methods, results and conclusions/research implications. In fact, I can not catch any effective information here.
  5. Why you emphasize Sodium adsorption ratio (SAR)? I do recognize its importance, however, you need to illustrate why you introduce it specifically?
  6. The uncertainty is another key issue in soil mapping, and the simulation with a low uncertainty is needed. The uncertainty is always evaluated by repeatedly conducting the simulations. In this study, the authors adopted machine-learning method, which may generate high uncertainty, and thus, the uncertainty evaluation is necessary. However, the authors even did not evaluate the uncertainty. The result generated by the simulation only at one time was adopted as the final result, which is inappropriate and considerably influences the reliability of the results.
  7. For 2. Theoretical background, it is crucial. The authors should make most efforts to clarify the estimating mechanism. In addition, you should also prove why the introduction of RS images improve the inversion. These problems are crucial in your study.

Specific comments:

  1. L31 R value at which level? 0.01 or 0.05?
  2. The introduction needs to be expanded.
  3. Please provide a bit more big-picture motivation of how your analyses benefit society and how they have evolved over the past decade. However, from my point of view, the article does not provide a sufficiently thorough review of the issue under study. There are good references for the study techniques, but the paper is missing a "big-picture" introduction with some references in my opinion. I suggest that the authors should do a better analysis of the literature. It seems that the bulk of the text is a sort of compilation of statements in the individual articles cited. It would be better, I think, to extract ideas from individual articles and tie them together into a more fluid and conceptually homogeneous text. As it is, the text looks rather clumsy.
  4. Research gaps, objectives of the proposed work should be clearly justified before the problem formulation section. This paper includes some little useful information and the main objectives of the study is not well defined. Problem statement is not clear and the objectives are obscure. Furthermore, the paper lacks a very clear and good justification for what is new and innovative about this case or this approach.
  5. For soil salinity inversion and uncertainty analysis, some recent articles available for similar studies can be read.—(1. Capability of Sentinel-2 MSI data for monitoring and mapping of soil salinity in dry and wet seasons in the Ebinur Lake region, Xinjiang, China[J]. Geoderma, 2019, 353: 172-187. 2. Machine learning-based detection of soil salinity in an arid desert region, Northwest China: A comparison between Landsat-8 OLI and Sentinel-2 MSI[J]. Science of the Total Environment, 2020, 707: 136092. 3. Ensemble machine-learning-based framework for estimating total nitrogen concentration in water using drone-borne hyperspectral imagery of emergent plants: A case study in an arid oasis, NW China[J]. Environmental Pollution, 2020, 266: 115412. 4. Updated soil salinity with fine spatial resolution and high accuracy: The synergy of Sentinel-2 MSI, environmental covariates and hybrid machine learning approaches[J]. CATENA, 2022, 212: 106054.)
  6. Explain how EC is determined and provide laboratory error.
  7. For quantifying study, only R2, RMSE, ME are not sufficient. Please use RPD or RPIQ. In fact, these two are more effective measures in remote sensing inversion.
  8. The detailed information soil sampling campaign and experimental design should be added.
  9. The authors should a related workflow diagram.
  10. The division of Sub-soil and top-soil should be added in your introduction.
  11. Table 1 is hard to follow. Please re-organize it. In addition, please add correct unit for each diagram.
  12. Why not consider the red edge information from Sentinel-2 for EC estimation?
  13. What platforms are the models and uncertainty analysis implemented on and which packages are employed? We encourage authors to include the code as supplementary documentation.
  14. In this study, NDVI and other related RS information seems to be not essential. Therefore, the authors should illustrate their motivation in details.
  15. Fig. 12 I can not get your aim. Please re-organize.
  16. Please re-write the conclusion in a concise manner (generally in one short paragraph).
  17. The Conclusions section should present a summary of the main findings and new knowledge, but not repeats of what have been discussed in the previous sections.

Reviewer 2 Report

 The term ECe is used differently than expected In Australia where the “e” is the texture which has an affect on the salinity.There is a reference which might be useful Hazelton and Murphy 2016.Use of the terms “higher “rather than “larger“

Reviewer 3 Report

This manuscript is very well-written and structured and has no major technical flaws. My main concern is that the number of soil observations is quite small and perhaps too small to do proper statistical inference and draw meaningful conclusions. I also wondered whether the manuscript has enough ‘remote sensing’ to justify publication in this journal. I concentrated my review on the methods since I am not an expert in soil electrical conductivity and its relation with other soil and environmental properties.

Detailed comments

L22 To me it would make more sense to write that ECa was 26% smaller in 2020 than in 2017.

L26-27 Why use firsts R-square and then R?

L89 “for assess” --> “to assess”

L113 The study period is from 2017 to 2019, so how can you report results for the year 2000? In fact L118 mentions February 2020.

L118-132 It was not clear to me how many ECa measurements were done in the field in 2017 and 2020. How dense was the spatial sampling and what was the sampling design? This is important because later on you use inverse distance interpolation to create a map, and interpolation error may be large in case of low sampling density.

L122-123 Why in this order, why not from 0.5 to 1 to 1.5 to 3 m?

L133-139 The meteorological conditions were not equal in 2017 and 2020, which is no surprise because it is impossible to select dates that have the same conditions. What effect does this have on ECa? For example, L22 mentions that ECa decreased from 2017 to 2020, but is this because there is a real decreasing trend or is it because the meteorological conditions between the two dates were different? I think this is important because it has a major effect on how results should be interpreted. In fact my question is: is it okay to measure on a single date only to characterise the system, or would a longer time series (say a period of one or two months) be needed in order to assess the trend and filter out day-to-day fluctuations caused by meteorological conditions?

L149-159 I found this text difficult, which may be because I am not an expert in this field. Still, it would be useful to explain more clearly how EC is derived from ECa (authors make clear in L73-74 that ECa is only a proxy of EC), how the differences in spatial support of the ECa-1.5, ECa-0.5, ECa-3 and ECa-1 measurements were handled, and what is the difference between 2D and 3D inversions ( I presume 2D refers to points along a transect, whereas 3D means a 2D-dimensional study area). I also wondered: were the inversions applied to the EMI signals or to the interpolated maps of the EMI signals?

L168 The Pearson correlation between interpolated ECa and EC may be different from the Pearson correlation between measured ECa and EC. How big is this difference? If I got it right then authors can check this because they derived EC from ECa at measurement locations in Section 2.2.1.

L157 This is the first time that mention is made of ‘the transect’, but what this transect is has not yet been explained (that is done later, in L174).

L169-171 What do you mean by spatiotemporal evolution, how is this quantified? And how was this ‘related’ with the estimated depth-specific ECe? Is ECe the same as EC? It gets quite confusing now. Authors must do a better job explaining the differences and relations between EC, ECa, ECe and SAR.

L173-182 Five locations is very limited, can we learn anything useful from such a small data set? Authors should make a convincing case that this is indeed sufficient. Also, the spatial support of the soil samples is much smaller (0.05 m diameter auger) than those of the electromagnetic measurements. Would it not have been better to take composite or bulk soil samples?

L193 Explain how the calibration was done: explain how many and which parameters of the inversion model were calibrated using the measured ECe and SAR of the soil samples. What calibration method was used, what penalty function was optimised? Note that there is a serious risk of overfitting since there were only 25 soil samples. Moreover, the high spatial clustering (5 soil samples at the same location at different depths) may lead to overoptimistic cross-validation results if LOOCV is used. It would be better to use leave-location-out cross-validation (i.e., using 20 soil observations from 4 locations to predict at the fifth location).

L198 Please provide the definition of the Coefficient of Determination. In fact I suggest that authors use the (Nash-Sutcliffe) Model Efficiency coefficient instead (https://en.wikipedia.org/wiki/Nash%E2%80%93Sutcliffe_model_efficiency_coefficient). Using the term ‘R-square’ is confusing because it can both refer to the R-square of the linear regression between predicted and observed (i.e., the squared Pearson correlation) and to the R-square with respect to the 1:1 line. The latter is what we want in case of cross-validation, and then to avoid confusion it is better to call it the model efficiency coefficient. Note also that RMSE and MAE are largely overlapping. Instead, it may be better to replace MAE by the Mean Error (ME). Although one should reasonably expect that the methods have no systematic error, it is always useful to do a check.

Table 1. All entries have one decimal point but that does not make sense, it is better to have all values have the same number of significant figures. For example, ECe might have two decimals while Na+ may be rounded to whole numbers. I also wondered whether it is useful to present the outcome of parametric statistical tests since the sample sizes are quite small.

Round 2

Reviewer 1 Report

Thank you for this revised paper. It is rather improved and ready to be published.